# Coupled Thermomechanical Response Measurement of Deformation of Nickel-Based Superalloys Using Full-Field Digital Image Correlation and Infrared Thermography

**DOI:** 10.3390/ma14092163

**Published:** 2021-04-23

**Authors:** Krzysztof Żaba, Tomasz Trzepieciński, Sandra Puchlerska, Piotr Noga, Maciej Balcerzak

**Affiliations:** 1Department of Metal Working and Physical Metallurgy of Non-Ferrous Metals, Faculty of Non-Ferrous Metals, AGH—University of Science and Technology, al. Adama Mickiewicza 30, 30-059 Cracow, Poland; spuchler@agh.edu.pl (S.P.); pionoga@agh.edu.pl (P.N.); maciejbalcerzak1@gmail.com (M.B.); 2Department of Materials Forming and Processing, Faculty of Mechanical Engineering and Aeronautics, Rzeszow University of Technology, al. Powst. Warszawy 8, 35-959 Rzeszów, Poland; tomtrz@prz.edu.pl

**Keywords:** digital image correlation, Inconel alloy, infrared mapping, nickel-based superalloys, temperature, thermovision, uniaxial tensile test

## Abstract

The paper is devoted to highlighting the potential application of the quantitative imaging technique through results associated with work hardening, strain rate and heat generated during elastic and plastic deformation. The aim of the research presented in this article is to determine the relationship between deformation in the uniaxial tensile test of samples made of 1-mm-thick nickel-based superalloys and their change in temperature during deformation. The relationship between yield stress and the Taylor–Quinney coefficient and their change with the strain rate were determined. The research material was 1-mm-thick sheets of three grades of Inconel alloys: 625 HX and 718. The Aramis (GOM GmbH, a company of the ZEISS Group) measurement system and high-sensitivity infrared thermal imaging camera were used for the tests. The uniaxial tensile tests were carried out at three different strain rates. A clear tendency to increase the sample temperature with an increase in the strain rate was observed. This conclusion applies to all materials and directions of sample cutting investigated with respect to the sheet-rolling direction. An almost linear correlation was found between the percent strain and the value of the maximum surface temperature of the specimens. The method used is helpful in assessing the extent of homogeneity of the strain and the material effort during its deformation based on the measurement of the surface temperature.

## 1. Introduction

While the elastic deformation of a metallic body resides in the reversible displacement of atoms from their equilibrium positions under the influence of applied forces, the induction of plastic deformation requires the activation of the phenomenon of dislocation glide and/or twinning. Both mentioned mechanisms are irreversible, i.e., after the disappearance of the load, the deformation of the material remains permanent [1]. Among the two plastic deformation mechanisms mentioned, dislocation glide is generally the dominant one. The stresses induced in twinning are greater than the stresses necessary for the motion in dislocation. For this reason, after the possibility of dislocation glide is stopped, plastic deformation can occur through twinning. Mechanical energy produced during plastic and elastic deformation of a metal is partially converted into heat while the rest is stored as deformation energy [2]. After loading, this stored energy remains in the metallic material as phase changes, permanent microstructural changes including internal defects formation [2,3]. The heat generated induces a temperature increase of the material. The plastic work *W_p_* is converted into heat *Q_p_* during the process of deformation. The ratio of plastic work converted into heat is defined by the coefficient *β*:(1)$$\beta =\frac{Q_{p}}{W_{p}}$$

Coefficient *β* is generally assumed as to be a constant in dependent of strain rate and plastic deformation and its value is commonly set as *β* = 0.9 [4]. Infrared techniques combined with a high-speed test allow one to estimate the coefficient *β.*

There are at least two main procedures for measuring the temperature increase asso-ciated with plastic strain. The first one is based on the measurement of the thermal radiation of the material emitted during plastic deformation. The second is based on the use of a rapid-response thermocouple connected to the test specimen. Infrared thermography has been widely used to correlate plastic strain to temperature increase [2]. Infrared radiation is emitted by any body with a temperature greater than absolute zero. The non-destructive nature of infrared mapping enables accurate measurement of the object’s temperature to be made in real time. Thermovision studies are possible due to the existence of infrared radiation, a type of electromagnetic radiation with a wavelength range from 0.78 µm to 1000 µm. Thermal radiation measurements are made in the far infrared range. The two main wavelength ranges are 3–5 µm and 8–13 µm.

Thermovision research, also known as infrared mapping, consists of the detection, registration and analysis of infrared radiation. These tests are a valuable tool for finding defects in the production processes of metals, ship skins [5] and painted sheet metals in the automotive industry [6]. Portable thermal imaging systems, such as cameras, are currently one of the best tools for measuring the temperature [7]. The result of thermographic research is a thermogram, a picture of the temperature field distribution on the surface of the test object. Infrared imaging can be divided into active and passive [8]. Active thermovision consists of supplying thermal energy to the test material. In response to the energy supplied, the material begins to emit infrared radiation, which is captured by a thermal imaging camera. Due to the supply of thermal energy, it is easy to notice temperature differences in the material and possible anomalies [9]. Passive thermovision is based on the measurement of the temperature field of an object that was created as a result of its operation [10,11].

During the two last decades, several authors have used infrared thermography to estimate the temperature increase of materials during deformation. Rusinek et al. [12] investigated the deformation process of dual phase steel at high strain rates. Rusinek et al. [4] conducted experiments on the heat generated during plastic deformation and stored energy for transformation-induced plasticity (TRIP) steels. Rodríguez-Martínez et al. [13] investigated the behaviour of mild steel sheets subjected to perforation by hemispherical projectiles using a high-speed infrared camera. Pérez-Castellnos and Rusinek [2] studied the temperature increase in 6068 aluminium alloy associated with plastic deformation induced under high strain rates. Thermal imaging studies are widely used in many industries. It is used in production lines, in construction diagnostics, electrical and mechanical engineering and medicine.

The Digital Image Correlation (DIC) technique coupled with infrared thermography provides a visualised measure to reveal the changes in the displacement field and temperature throughout the tensile tests. Coupled experiments using DIC and infrared thermography were used by Feng and Xue [14] to study the deformation of 3D printed bolts of different materials. Acciaioli et al. [15] investigated the suitability of DIC for measuring homogeneous strain field, by experimentally determining the accuracy achievable using DIC to measure the Poisson’s ratio and Young’s modulus of a specimen subjected to strain levels smaller than 0.1%. Wu et al. [16] concluded that DIC systems can measure high strain levels with small errors. Strain errors decrease if the specimen’s dimensions increase [17]. The DIC technique is accurate in measuring small, lower that 0.1% [18,19] and medium–large strain levels [20,21,22,23]. Gamboa et al. [24] applied the digital image correlation technique to the study of material flow under 2D and 3D conditions that take place during an indentation process. Cholewa et al. [25] introduced a new fused infrared thermography DIC (TDIC) technique applicable to small to large length-scales. This technique is demonstrated through measurements obtained on a sandwich composite subjected to simultaneous one-sided heating and compressive loading resulting in non-uniform temperatures. The thermal field is commonly measured, assuming a sufficiently thin specimen, relative to its thermal conductivity such that a negligible through-thickness thermal gradient exists [26]. Thus, DIC and infrared thermography measurements may be performed on opposing surfaces [26,27]. Maynadier et al. [28] proposed a simultaneous infrared thermography and DIC system that used a high emissivity speckle pattern to measure deformation and thermal fields using a single infrared camera. This technique required the use of a special coating that may be discerned as a speckle pattern. Chrysochoos et al. [26] used DIC and infrared thermography to study heat sources and mechanical energy during a heterogeneous tensile test. The results approached the concepts of the scale dependence of hardening/softening behaviour, elastic threshold and plastic plateau, and the existence of mechanical hysteresis loops induced by coupling mechanisms and heat diffusion. It was proven that the displacement and strain maps of a structure would often be difficult to interpret without data provided by passive thermography about the emissivity of the object being investigated [29]. A procedure for multimodal 3D DIC-infrared thermography measurements was proposed by Rutkowski et al. [30]. A few examples and future applications have been shown. In the case of thin, flat objects, DIC and infrared thermography systems are placed on opposite sides of an object [26]. Otherwise, a mirror and a cold black body are applied to provide measurements using a single infrared camera.

Most of the research is focused on the analysis of the effect of strain on the temperature of material during high strain rates in the range between 100 and 6000 s^−1^. A little less was done for the range of strain rates achievable on universal tensile testing machines. Moreover, to the best of the authors’ knowledge confirmed by Dolci et al. [31] no published data is available for the Taylor–Quinney coefficient for Inconel-718. This coefficient has been determined for the analysis of the range of strain rates of 0.004–0.016 s^−1^. This paper presents coupled thermo-mechanical response measurements of the deformation of Ni-based superalloys using full-field DIC and infrared thermography. Three grades of Inconel alloy sheets, i.e., 625 (AMS 5599), HX (AMS 5536) and 718 (AMS 5596), were tested. The dog-bone test specimens were cut along three directions from 1-mm-thick sheets at an angle of 0°, 45° and 90° with respect to the rolling direction of the sheet metal. Uniaxial tensile tests were carried out at three different speeds. The proposed method may be used for the analysis of sheet metal forming of Ni-based materials to detect or to control the locations in the drawpiece that are prone to cracking. Measurement of sheet temperature is much easier than measurement of deformation. Such a strategy must be preceded by the development of a methodology for the correlation of temperature value with the value of sheet deformation, which is the subject of this study.

## 2. Materials and Methods

### 2.1. Material

In the investigations of the effect of strain on the temperature of the material subjected to uniaxial tensile testing, three grades of Ni-based superalloys were used, i.e., Inconel alloy 625 (AMS 5599), Inconel alloy HX (AMS 5536) and Inconel alloy 718 (AMS 5596). The dog-bone specimens (Figure 1) were cut from 1-mm-thick sheets at an angle of 0°, 45° and 90° with respect to the rolling direction of the sheet metal. Inconel 625 is a nickel-chromium-molybdenum alloy that is used for its high strength, excellent weldability and fabricability, and outstanding long-term corrosion resistance. Inconel 625 has been the most widely used alloy in aircraft exhaust systems, vector nozzles and tailpipes in the aerospace industry. Typical marine applications of Inconel 625 include propulsion motors, steam liner bellows and quick-disconnect fittings. Inconel alloy HX is a matrix-stiffened, high-temperature, nickel–chromium–iron–molybdenum alloy with outstanding oxidation resistance and good resistance to chloride stress-corrosion cracking. It is typically used for components like combustion chamber afterburners, tail pipes in aircraft and land-based gas turbine engines. Inconel 718 is high-strength corrosion-resistant nickel-chromium material that is highly resistant to corrosion in oxidising environments. This alloy is used for various sheet metal parts for aircraft and land-based gas turbine engines, casings and fuelled rockets. The chemical composition of the tested sheets are shown in Table 1. For brevity, the names of the alloys tested were shortened as I625 (Inconel 625), HX (Inconel HX) and I718 (Inconel 718).

### 2.2. Uniaxial Tensile Test

The specimens were stretched in a Zwick/Roell Z100 (Zwick/Roell, Ulm, Germany) uniaxial tensile testing machine. The tests were carried out at three different strain rates: 0.004 s^−1^, 0.008 s^−1^ and 0.016 s^−1^.

### 2.3. Digital Image Correlation Technique

A material-independent measuring system based on DIC Aramis (GOM, Braunschweig, Germany) was used for non-contact analysis of sample deformation during the uniaxial tensile tests. Aramis is a system for three-dimensional strain measurements under both static and dynamic loads. Graphical presentation of measurement results in the form of strain distributions gives the possibility of a more complete understanding of the behaviour of the test object. Aramis recognises the three-dimensional structure of the surface of the object being measured on the basis of photos taken with a digital camera (each pixel in the image being assigned appropriate coordinates).

The samples were prepared by applying a random pattern to their surface in the form of black speckles on a white background. The surface of the sheet metal with a pattern is scanned with a camera with a certain resolution. As a result, the digital image of the scanned area has a certain number of pixels. The measuring principle is based on recording successive frames while the material is deforming. When a series of images is taken, the subsequent stages are compared to a reference (undeformed) stage and full-field strains can be calculated. The displacement of a point is the difference between the original position of the point and its displacement in which the position shifted is the one in which the correlation function representing the degree of grey in the vicinity of the point on the body surface reaches an extreme [35]. The DIC approach therefore provides global information about changes to the structure over the entire field of view captured in pictures. Deformations are calculated from the image stages by extracting a set of 3D points from the surface and applying finite-strain equations. The continuum mechanics-based finite-strain equations with original coordinates *x*, *y*, *z* and displacements *u*, *v*, and *w* are:(2)$$\varepsilon_{x}=\frac{\partial u}{\partial x}+\frac{1}{2}\left[{\left(\frac{\partial u}{\partial x}\right)}^{2}+{\left(\frac{\partial v}{\partial x}\right)}^{2}+{\left(\frac{\partial w}{\partial x}\right)}^{2}\right]\;\varepsilon_{y}=\frac{\partial v}{\partial y}+\frac{1}{2}\left[{\left(\frac{\partial u}{\partial y}\right)}^{2}+{\left(\frac{\partial v}{\partial y}\right)}^{2}+{\left(\frac{\partial w}{\partial y}\right)}^{2}\right]\;\varepsilon_{xy}=\frac{1}{2}\left[\frac{\partial u}{\partial x}+\frac{\partial v}{\partial x}+\left(\frac{\partial u}{\partial x}\frac{\partial u}{\partial y}\right)+\left(\frac{\partial v}{\partial x}\frac{\partial v}{\partial y}\right)+\left(\frac{\partial w}{\partial x}\frac{\partial w}{\partial y}\right)\right]$$

The calibration of the scanner involved: (i) measuring the distance the scanner should be from the specimen, (ii) setting the scanner lamps in such a way that the sample was not overexposed, and (iii) taking a series of photos using a special calibration panel. The Aramis system has many advantages. It allows a researcher to measure objects of various sizes, from 1 mm to 2 m, with the same sensor. Relative strains can be measured in the range between 0.01% to several hundred percent.

### 2.4. Infrared Thermal Mapping

The temperature distribution in the specimen during its deformation was measured with an FLIR T640 (FLIR, Wilsonville, OR, USA) high-sensitivity infrared thermal imaging camera with ResearchIR software (version 4.40.11.35, FLIR Systems, Wilsonville, OR, USA). A thermal imaging camera allows one to obtain an image of the temperature distribution by means of a non-invasive distance measurement, which allows the researcher to read the electromagnetic emission of waves invisible to the eye in the spectral range from 7.5 to 13 μm. The parameters of the camera are: detector resolution 640 × 480, thermal sensitivity <30 mK at 30 °C, temperature range from −40 °C to +2000 °C, refresh rate 30 Hz, measurement accuracy: ±2 °C, field of view 25° × 19°/0.25 m, 5 Mpix camera (2048 × 1536 pixels). A picture of the test stand is presented in Figure 2.

## 3. Results and Discussion

Figure 3, Figure 4 and Figure 5 show the DIC cloud charts for strain fields and corresponding infrared thermography images for I625 alloy. During the testing of I625 sheet metal at a strain rate of 0.004 s^−1^, a clear oval distribution of temperature was observed on the surface of the specimens. The area with the highest temperature was located in the vicinity of the plane of longitudinal symmetry of the specimen (Figure 3).

There is a clear gradient of temperature changes towards the edge of the specimen. Increasing the strain rate to 0.008 s^−1^ made the distribution of temperature change over the width of the sample more uniform (Figure 4). While for the strain rate of 0.008 s^−1^ there is a correlation between the location of the necking and the area of the highest temperature (Figure 4), at a strain rate of 0.004 s^−1^, the area of highest temperature clearly exceeds the area of necking corresponding to the strain ε = 77.82% (Figure 3). After the onset of necking, plastic strain was localised in the necked region, whereas the plastic deformation will remain constant outside the neck [31]. After necking, the strain rate was increased in the necked area and was not uniform over the specimen. Therefore only the material in the vicinity of the necking undergoes further deformation, while the material elsewhere remains at constant plastic strain.

There is also a clear tendency for sample temperature to increase with an increase in the strain rate. In general, similar general conclusions can be applied to the other two sheets, I718 and HX, and all directions of sample cutting. The influence of specimen orientation on the temperature distribution of the specimens made of the I718 sheet is shown in Figure 6, Figure 7 and Figure 8.

Figure 9, Figure 10 and Figure 11 show infrared tomography images of the influence of sheet-rolling direction and strain rate on the temperature distribution of HX specimens. It is well known that in the range of proportional plastic strains, the cross-section of the specimen varies uniformly over its entire length of measurement. This phenomenon is confirmed by the recorded temperature distributions, since a uniform temperature distribution occurs along the gauge length of a test piece. In general, the locations of loss of stability do not always coincide with the area of the highest temperature, which occurs in the middle part of the sample being measured.

There is an almost linear correlation between the percentage strain and the value of the maximum temperature on the surface of the specimens. For most specimens, the linear correlation between these parameters, determined by the determination coefficient R^2^, is greater than 0.99. Specimens from the I625 sheet, cut in different directions and tested at all the strain rates, show an almost identical character (Figure 12). It is an indicator of the isotropic properties of the sheet.

There is a distinct change in the angle of slope of the trend line in relation to the abscissa. The higher the strain rate, the steeper the trend line (the greater the slope of the straight line). Clear anisotropic features are shown by specimens made of sheets of I718 (Figure 13) and HX (Figure 14). The elongation was highest at the necking of the HX alloy specimens cut at an angle of 45°. An increase in the maximum temperature of the samples cut at this angle is also clear. When specimens made of I718 and HX alloys were tested at a strain rate of 0.008 s^−1^, the maximum deformation was similar at cut directions of 0° and 90°.

The difference in the maximum temperature of the specimens made of I625 tested at 0.004 s^−1^ at the moment of necking differed by about 29.9% between the orientation of the samples at 0° and 90° (Figure 12a). Increasing the strain rate to 0.008 s^−1^ reduced this difference to 4.3% (Figure 12b). The same difference between the extreme temperatures of the sheet surface (Figure 13) at the moment of necking when measured in I718 alloy was 14.7% (strain rate 0.004 s^−1^), 22.6% (strain rate 0.008 s^−1^) and 9.6% (strain rate 0.016 s^−1^).

Tensile stress–strain curves for the test material are presented in Figure 15, Figure 16 and Figure 17.

In the case of the HX (Figure 15) and I625 (Figure 16) alloys, after increasing the strain rate from 0.004 s^−1^ to 0.016 s^−1^, it was observed that the material showed greater isotropic properties. The effect of work hardening on the value of stress in the range of uniform deformation is very similar for all sample orientations, while for the 0.004 s^−1^ (Figure 15c) and 0.008 s^−1^ (Figure 16c) strain rates, there is a clear difference in the course of the tensile curves and the strain values corresponding to the sample breaking. The difference in elongation at the break value of the samples tested at 0.016 s^−1^ is 2.9% and 5.3% for the HX alloy samples (Figure 15c) and I625 (Figure 16c), respectively. As for the HX and I625 alloys, the difference in elongation at the break for I718 alloy was the smallest at the highest strain rate (Figure 17c). There are no clear dependencies of the influence of the sample orientation on the value of the stress in the sample in the range of plastic deformation.

The yield stress determined at 0.2% plastic elongation (R_p0.2_) increased with the increase in the strain rate for all materials and orientations (Figure 18). An approximation of the experimental data of the HX alloy (Figure 18a) using linear regression led to the conclusion that the slowest increase in yield stress value with an increase in the strain rate occurred for the orientation 0°. In contrast, the greatest value of yield stress is observed for samples cut towards 45°. Sheets made of I625 alloy (Figure 18b) were characterised by the greatest anisotropy in relation to the value of yield stress. The difference in the yield stress for a specific strain rate for specimens cut along the sheet-rolling direction (0°) and with a specimen orientation 45° was in the range between 6 and 9 MPa. Sheets made of the I718 alloy showed the closest values of yield stress among all the orientations tested (Figure 18c). If the strain rate increases the difference between yield stresses for different orientations decreases.

The correlations between the variation of the yield stress and measured temperature for the test materials is shown in Figure 19, Figure 20 and Figure 21.

For the HX alloy, an upward trend of changes in values of both yield stress and temperature is visible with the increase of the strain rate. The type of correlation is quite difficult to interpret. While for orientation 45° (Figure 19b) and 90° (Figure 19c) the linear correlation between yield stress and strain rate is quite strong (R^2^ > 0.82), for orientation 0° (Figure 19a) the linear correlation is rather excluded (R^2^ < 0.4). I625 alloy is characterised by a very small increase in yield stress value along with an increase in the strain rate from 0.004 s^−1^ to 0.016 s^−1^: 0.9% (Figure 20a), 1.0% (Figure 20b) and 2.1% (Figure 20c). At the same time, there is a clear linear trend towards an increase in temperature (R^2^ > 0.68). The value of yield stress for I718 alloy is linearly correlated with the increase of strain rate with coefficient of determination R^2^ > 0.98 (Figure 21). However, at the same time the temperature variation is not linearly correlated (R^2^ < 0.5) with an increase in the strain rate.

The Taylor–Quinney coefficient specifies fraction of plastic work converted to heat. According to the Zehnder model [31] the coefficient *β* is related to the calculation of stored energy per unit of dislocation density [2]:(3)$$\left({\bar{\varepsilon }}^{p}\right)=1-\frac{\partial \bar{\sigma }\left({\bar{\varepsilon }}^{p}\right)}{\partial {\bar{\varepsilon }}^{p}}{\frac{1}{E}|}_{{\dot{\bar{\varepsilon }}}^{p}}$$

For the work hardened Inconel alloy, the Taylor–Quinney (T-Q) coefficient under an assumption of isotropic hardening can be defined as [36]:(4)$$\beta \left({\bar{\varepsilon }}^{p}\right)=1-n{\left(\frac{{\bar{\varepsilon }}^{p}}{\varepsilon_{0}}\right)}^{n-1}$$
where *n* is the hardening exponent of the material, *ε*_0_ is the yield strain, ${\bar{\varepsilon }}^{p}$ is the plastic strain.

Figure 22, Figure 23 and Figure 24 show the variation of the Taylor–Quinney coefficient determined from Equation (4) with the increase of true strain. In the range of true strain from 0 to 0.05 a clear increase in the value of the T-Q coefficient is observed from a value of about 0.7. In this range of deformations, the clear influence of strain rate on the value of the T-Q coefficient is also visible. The rapid increase in the value of the T-Q coefficient right after reaching the yield stress is related to the large slope of the stress increment curve in this range (Figure 15, Figure 16 and Figure 17). The average value of the T-Q coefficient for all test materials and strain rates was found to be equal to 0.99. There is a direct correlation of the value of the T-Q coefficient with the value of the strain hardening exponent. The lower the value of the strain hardening exponent, the greater is the value of the T-Q coefficient. The greater the true strain, the more the value of the T-Q coefficient asymptotically approaches 1 and the difference in the value of the T-Q coefficient determined at different strain rates becomes the same.

## 4. Discussion

The plastic deformation of Ni-based alloys at low strain rates leads to an increase in the temperature of the material due to thermomechanical coupling. As mechanical work is done on the material, a portion of the work is spent deforming the microstructure and increasing the latent energy while a larger portion is dissipated as heat to the surroundings [37]. It was found that stretching the dog-bone samples made of three Ni-based alloys causes the material temperature to rise to about 60–90 °C (measured at the moment of necking) depending on the alloy grade and strain rate (Figure 3, Figure 4, Figure 5, Figure 6, Figure 7, Figure 8, Figure 9, Figure 10 and Figure 11). As a rule, no differences in the nature of temperature changes were found for different orientations during the uniaxial tensile test (Figure 6, Figure 7 and Figure 8). However, the location where sample stability is lost near the grippers does not coincide with the area of highest temperature in the preceding phase of the stretching process. Therefore, it is difficult to predict the location of sample destruction on the basis of the initial temperature distribution. The formation of necking at the moment of the material’s loss of stability is a complex and often sudden phenomenon. After necking, a large amount of heat is generated due to large plastic deformations [38]. However low strain rates soften the path-rerouting constraints thus only a small amount of energy is converted to heat [39]. With the beginning of the necking, the maximum temperatures and plastic strains gradually concentrate on the necking zone.

With increasing strain rates, the maximum temperature in the material increases (Figure 3, Figure 4, Figure 5, Figure 6, Figure 7, Figure 8, Figure 9, Figure 10 and Figure 11). Increasing the strain rates in tension not only results in strain hardening but also an increase in dislocation density [40,41]. A large number of dislocation pairs occurring after a few percent deformation is associated with γ″ particles. The primary strengthening phase γ″ precipitate coherently as disc-shaped particles and are based on Ni_3_Nb [42]. Pairs of dislocations in monotonically strained I718, as observed by Oblak et al. [41], are evidence of shearing of the γ″ phase by glide dislocations. The shearing of the γ″ phase within the matrix could occur due to extensive work hardening in the slip bands. The increased internal resistance due to dislocation tangling within the slip bands leads to an increase in plastic-deformation energy, which is converted into heat [42].

The resistance to plastic deformation is a rate-controlling process and can be affected by the strain rate [43]. The yield stress determined at 0.2% plastic elongation (R_p0.2_) increased with an increase in the strain rate for all materials and orientations (Figure 18). A linear effect of strain rate on yield stress was observed (Figure 19, Figure 20 and Figure 21), which is consistent with the results of Yan and Xu [44]. The positive strain rate sensitivity for yield stress could be due to the higher flow stress which is achieved at a higher strain rate since a higher strain rate might decelerate the dislocation annihilation [45]. Moreover, the higher strain rate increases the barriers of dislocation motion for the mechanically and thermally activated plastic deformations [46].

In the case of Inconel alloy I718 sheets, the location of necking near the gripping parts of specimen does not coincide with the area of the highest temperature in the preceding phase of the tensile process. Therefore, it is difficult to predict the location of sample destruction on the basis of the initial temperature distributions. Part of the mechanical energy created during a plastic deformation process in metals is converted to heat, while the remainder is stored in the material microstructure [47]. This part of the energy represents the change in internal energy of the metallic materials. It is commonly known that most of the mechanical energy is dissipated as heat during plastic strain. Understanding the relationship between heat evolution and the work of deformation is key to predicting temperature fields. In high-strain-rate deformations of Inconel alloys, heat is associated with thermal softening.

In the case of high rates of deformation, a gradual nonlinear increase in yield stress, which is known as strain rate sensitivity, is observed. At low rates of deformation, there is a linear dependence of the yield stress and the strain rate while at high rates of deformation, a rapid, non-linear increase in stress occurs with an increase in the strain rate [44,48]. According to the Peierls–Nabarro theory, there is a long-standing contradiction that the stable configuration of dislocations has maximum energy rather than minimum energy [49]. As temperature increases, the Peierls–Nabarro stress value decreases, which causes a slip within the plane of the atoms of a unit cell. This slide is limited by the intersecting atomic planes of the unit cell with dislocation lines generated by Frank–Read sources [50]. In technical alloys, dislocations moving in intersecting planes interact with each other and increase their density, leading to its strengthening. This phenomenon is related to the natural tendency of dislocation loops to expand and contract [50]. As the alloy temperature increases, the grain boundaries may limit the brittleness of the material and may be locations for the accumulation of dislocations. The areas of intense deformation within the grains move to the zones near the grain boundaries at higher temperatures [48].

The driving forces of work hardening of Inconel alloys, which is non-linear [51,52], are the dislocation glide and twinning mechanisms. As a result, plastic-deformation energy is converted into heat, the value of which and possible dissipation depends on the strain rate. The thermal-balance equation for isotropic material is provided by Rusinek and Klepaczko [3] as follows:(5)$$\lambda \nabla^{2}T-\dot{T}=-\frac{\beta }{\rho C_{p}}\sigma :\;{\dot{\varepsilon }}^{p}+\frac{\alpha }{\rho C_{p}}\frac{E}{1-2\nu }T\;tr\left({\dot{\varepsilon }}^{e}\right)$$
where *T* is the absolute temperature, *λ* is the thermal diffusitivity of the material, *ρ* is the mass density, *β* is the Taylor–Quinney coefficient [53] as a proportion of the plastic-deformation energy converted into heat, *σ* is the stress tensor, *C_p_* is the specific heat, ${\dot{\varepsilon }}^{p}$ is the plastic strain-rate tensor, α is the thermal-expansion coefficient, ν is Poisson’s ratio, E is Young’s modulus and ${\dot{\varepsilon }}^{e}$ is the elastic-strain-rate tensor.

If thermoelasticity is not considered, then the temperature increment ΔT may be calculated according to the formulae [3]:(6)$$\Delta T\left({\bar{\varepsilon }}^{p}\right)=\int_{0}^{{\bar{\varepsilon }}_{max}^{p}}\frac{\beta }{\rho C_{p}}\bar{\sigma }\left({\bar{\varepsilon }}^{p}\right)d{\bar{\varepsilon }}^{p}$$
where ${\bar{\varepsilon }}^{p}$ is the equivalent plastic strain and $\bar{\sigma }$ is the corresponding equivalent stress under uniaxial deformation.

Although the *β* coefficient is typically assumed to be constant (*β* = 0.9) [13,47], a different model has been developed to calculate the plastic-strain dependent Taylor–Quinney coefficient *β* [36,54,55]. No published data is available for the Taylor–Quinney coefficient for Inconel-718 [31]. Dolci et al. [31] found a constant value of *β* = 0.8, which best matched the stress–strain test behaviour exhibited in the tension test of I718.

The Taylor–Quinney coefficient expresses the efficiency of the thermomechanical conversion of the thermal dissipation to mechanical work involved in the deformation process. It was found that for the tested Ni-based alloys, the value of the T-Q coefficient calculated during homogeneous deformation in the sample before necking increases with the increase of plastic strain (Figure 22, Figure 23 and Figure 24). This relationship is in agreement with the results of the work of Smith et al. [37]. The average value of the T-Q coefficient of 0.99 determined from Figure 22, Figure 23 and Figure 24 is quite similar to the universal value of 0.9 commonly assumed for all metallic materials [56]. The effect of a strain rate in the range between 0.004 s^−1^ and 0.016 s^−1^ on the value of the T-Q coefficient is not clearly revealed. As reported by Zubelewicz [39] for strain rates not exceeding 100 s^−1^, the Taylor–Quinney coefficient is mildly sensitive to the rate of loading and the sensitivity becomes negligible in quasi-static conditions. Therefore the strain rate sensitivity is assumed to be an irrelevant factor at strain rates below 50 s^−1^.

The results of investigations show the potential of coupled full-field DIC and infrared tomography techniques to determine the cold deformation values of Inconel alloys based on the measurement of temperature. The method proposed may be useful in the thermomechanical analysis of the sheet metal forming of Ni-based alloys. The strategy may be extended to other metallic materials based on the correlation of the temperature values with the strains in sheet metal found using the proposed coupled thermomechanical response measurement of deformation using full-field DIC and infrared thermography.

## 5. Conclusions

In this paper, an experimental analysis is presented describing the effect of strain rate and strain value on the deformation-induced temperature of specimens made of three kinds of Nickel-based superalloys, i.e., Inconel alloy 625 (AMS 5599), Inconel alloy HX (AMS 5536) and Inconel alloy 718 (AMS 5596) subjected to plastic deformation in uniaxial tensile stresses. The following conclusions are drawn from the analysis and DIC cloud charts and they are concordant with the results of infrared thermography images:Although the deformation-induced increase in temperature is well known, there is also a clear tendency to increase the sample temperature with an increase in the strain rate. This conclusion applies to all materials analysed and all directions of sample cutting with respect to the sheet-rolling direction.In the range of plastic strains preceding necking, the highest temperature occurs in the middle part of the specimen.There is an almost linear correlation between the percent strain and the value of the maximum surface temperature of the specimens.Inconel I625 exhibits isotropic properties—the relationship of temperature versus strain overlaps for all specimen orientations.The yield stress, determined at 0.2% plastic elongation, increased with the increase in the strain rate for all materials and orientations.The value of the T-Q coefficient is calculated during homogeneous plastic deformation in the sample before necking increases with the increase of plastic strain.

## Figures and Tables

**Figure 1 materials-14-02163-f001:**
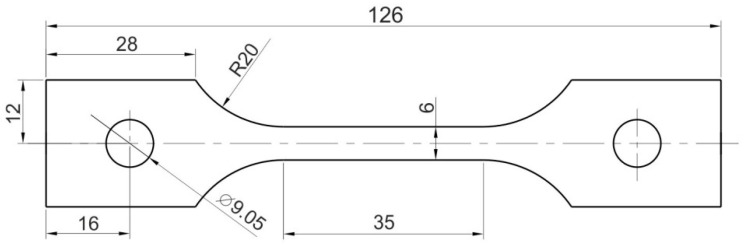
Dimensions (in mm) of the specimens for uniaxial tensile test.

**Figure 2 materials-14-02163-f002:**
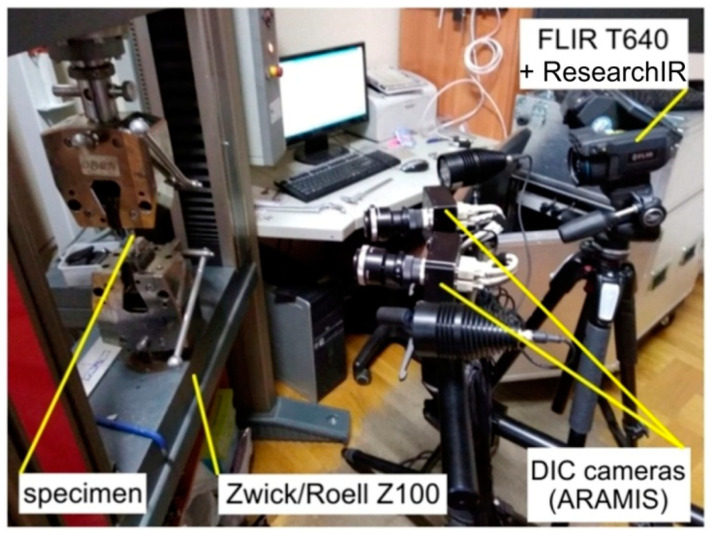
Test stand.

**Figure 3 materials-14-02163-f003:**
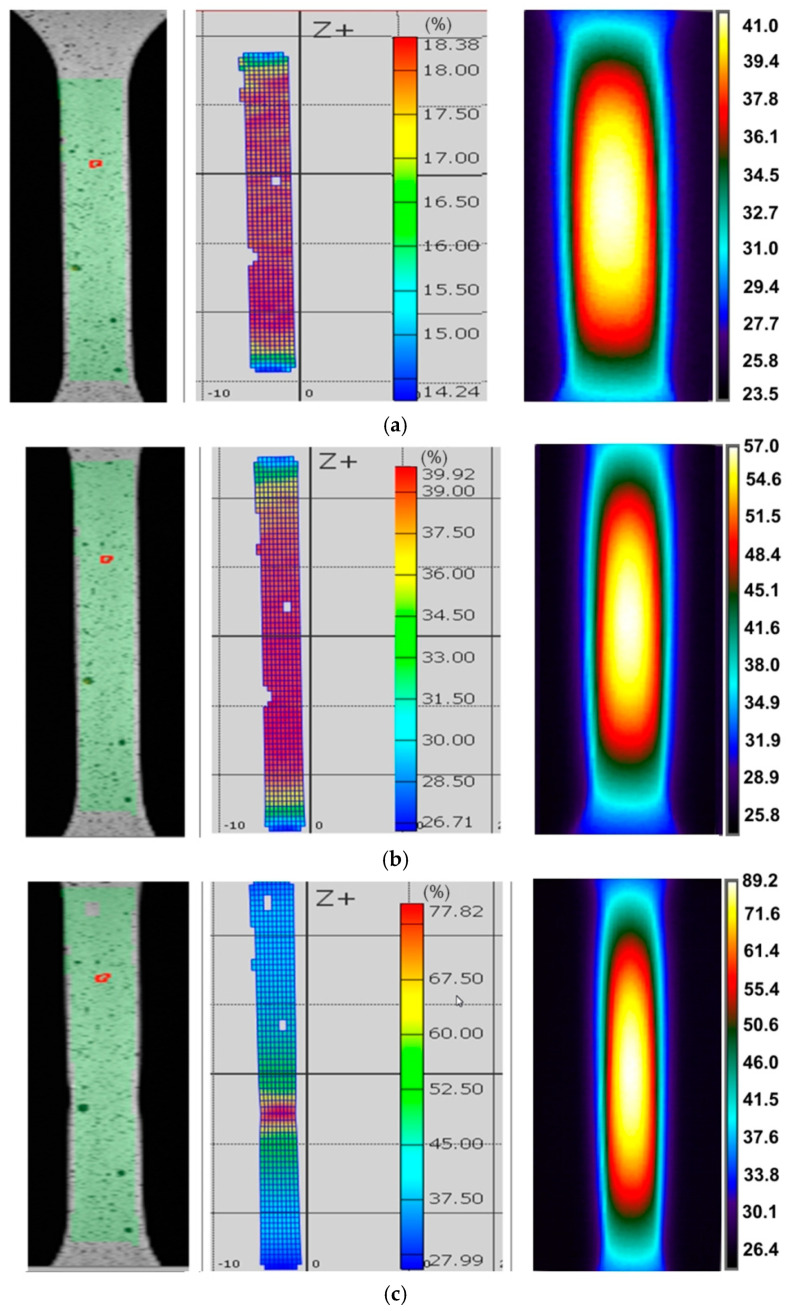
DIC cloud charts for strain fields and infrared thermography images for I625 specimens cut at an angle of 45° with respect to the rolling direction of the sheet metal, strain rate 0.004 s^−1^ and strains: (**a**) 18.38%, (**b**) 39.92% and (**c**) 77.82%.

**Figure 4 materials-14-02163-f004:**
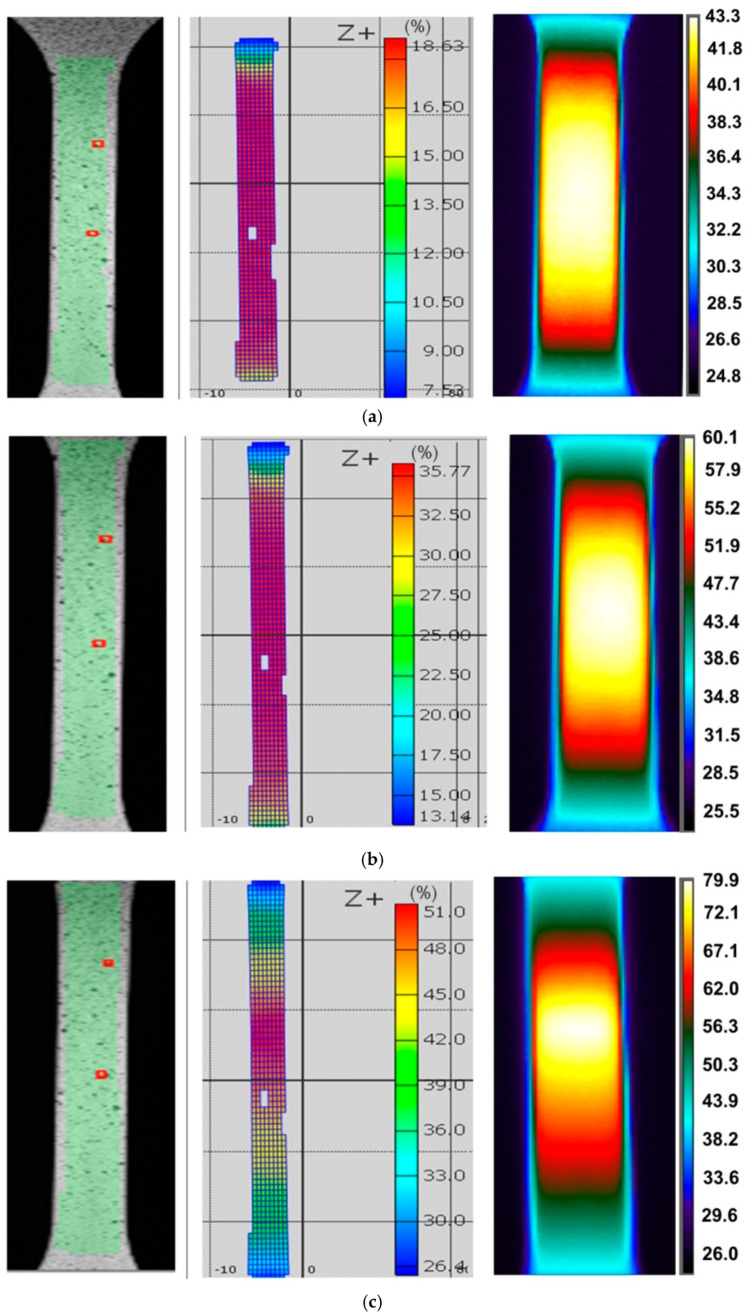
DIC cloud charts for strain fields and infrared thermography images for I625 specimens cut at an angle of 45° with respect to the rolling direction of the sheet metal, strain rate 0.008 s^−1^ and strains: (**a**) 18.63%, (**b**) 35.77% and (**c**) 51.10%.

**Figure 5 materials-14-02163-f005:**
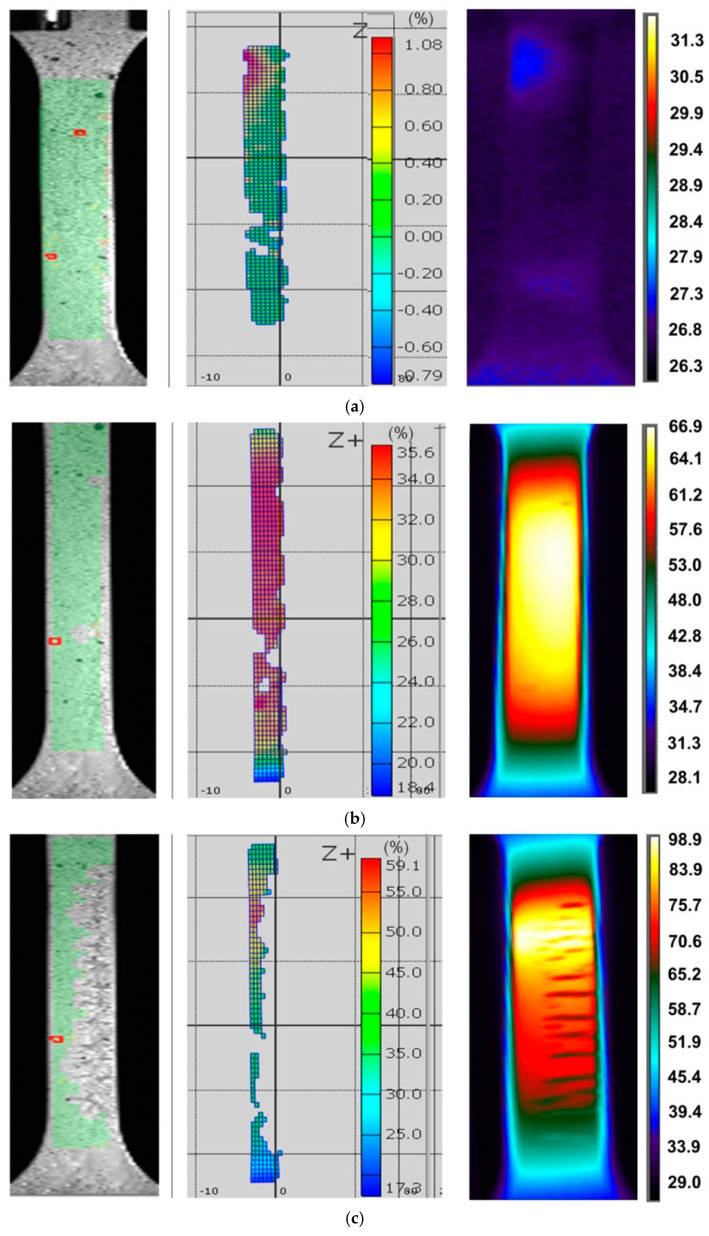
DIC cloud charts for strain fields and infrared thermography images for I625 specimens cut at an angle of 45° with respect to the rolling direction of the sheet metal, strain rates 0.016 s^−1^ and strains: (**a**) 1.08%, (**b**) 35.6% and (**c**) 59.1%.

**Figure 6 materials-14-02163-f006:**
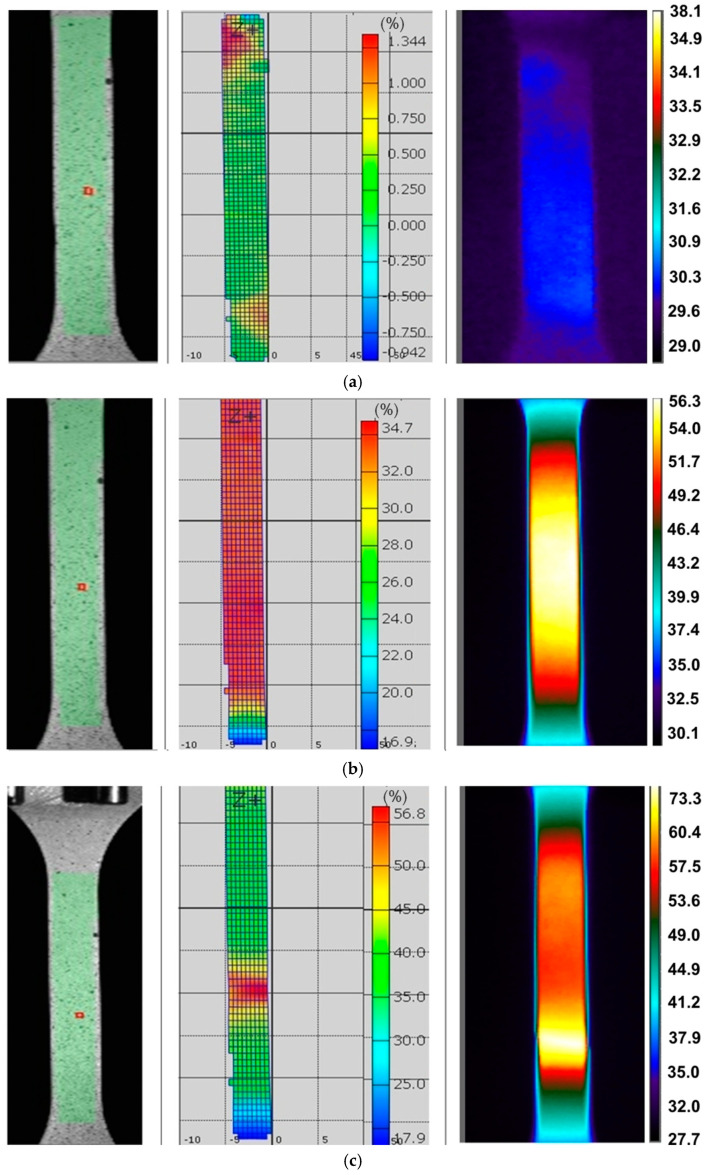
DIC cloud charts for strain fields and infrared thermography images for I718 specimens cut at an angle of (**a**) 0° with respect to the rolling direction of the sheet metal, strain rate of 0.004 s^−1^ and strains: (**a**) 1.34%, (**b**) 34.7% and (**c**) 56.8%.

**Figure 7 materials-14-02163-f007:**
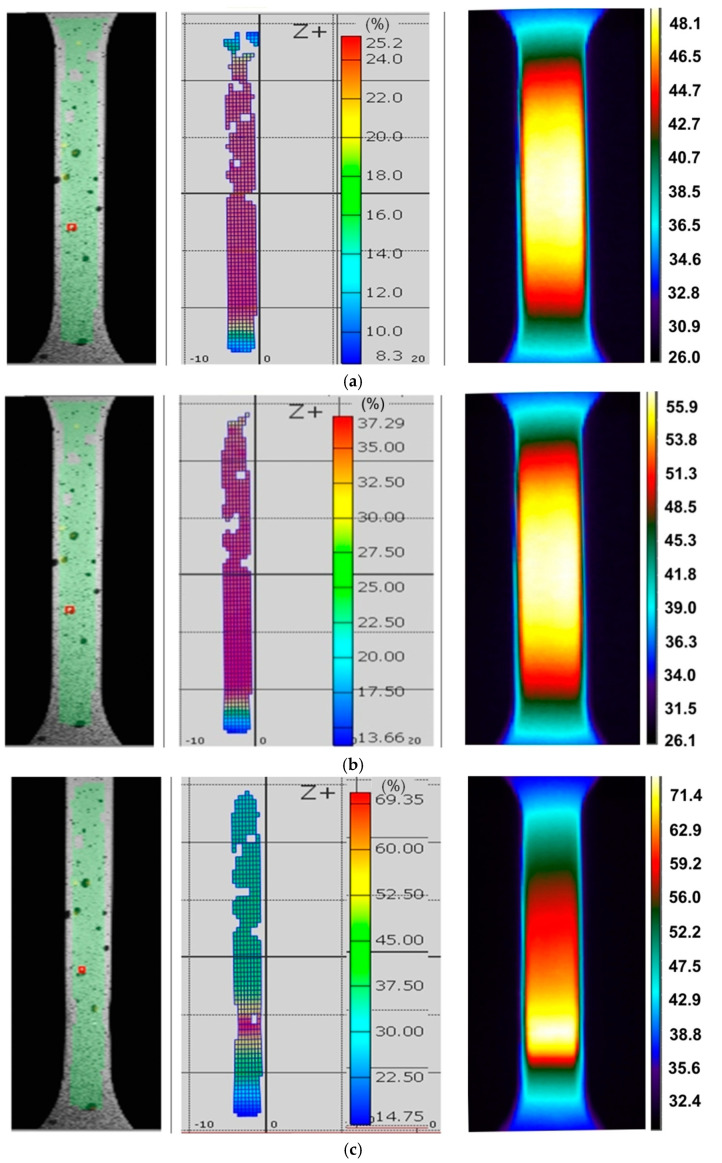
DIC cloud charts for strain fields and infrared thermography images for I718 specimens cut at an angle of 45° with respect to the rolling direction of the sheet metal, strain rate of 0.004 s^−1^ and strains: (**a**) 25.2%, (**b**) 37.28% and (**c**) 69.35%.

**Figure 8 materials-14-02163-f008:**
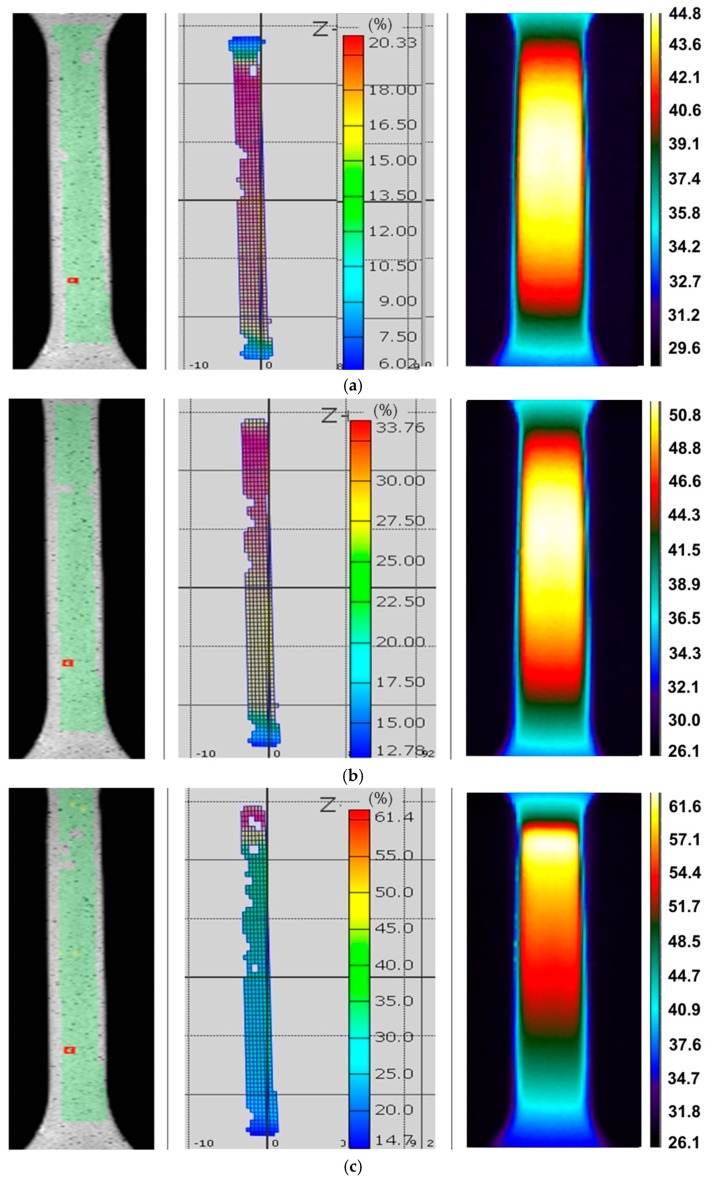
DIC cloud charts for strain fields and infrared thermography images for I718 specimens cut at an angle of 90° with respect to the rolling direction of the sheet metal, strain rate of 0.004 s^−1^ and strains: (**a**) 20.33%, (**b**) 33.75% and (**c**) 61.4%.

**Figure 9 materials-14-02163-f009:**
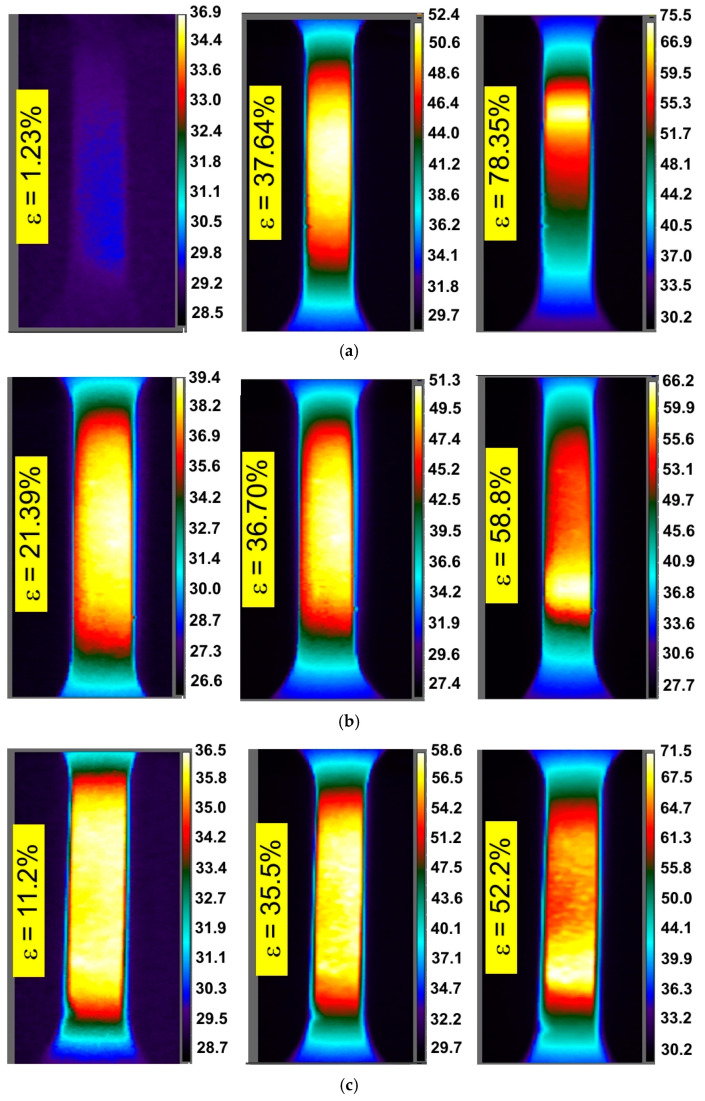
Infrared thermography images for HX specimens cut parallel to the rolling direction of the sheet metal tested at strain rates: (**a**) 0.004 s^−1^, (**b**) 0.008 s^−1^ and (**c**) 0.016 s^−1^.

**Figure 10 materials-14-02163-f010:**
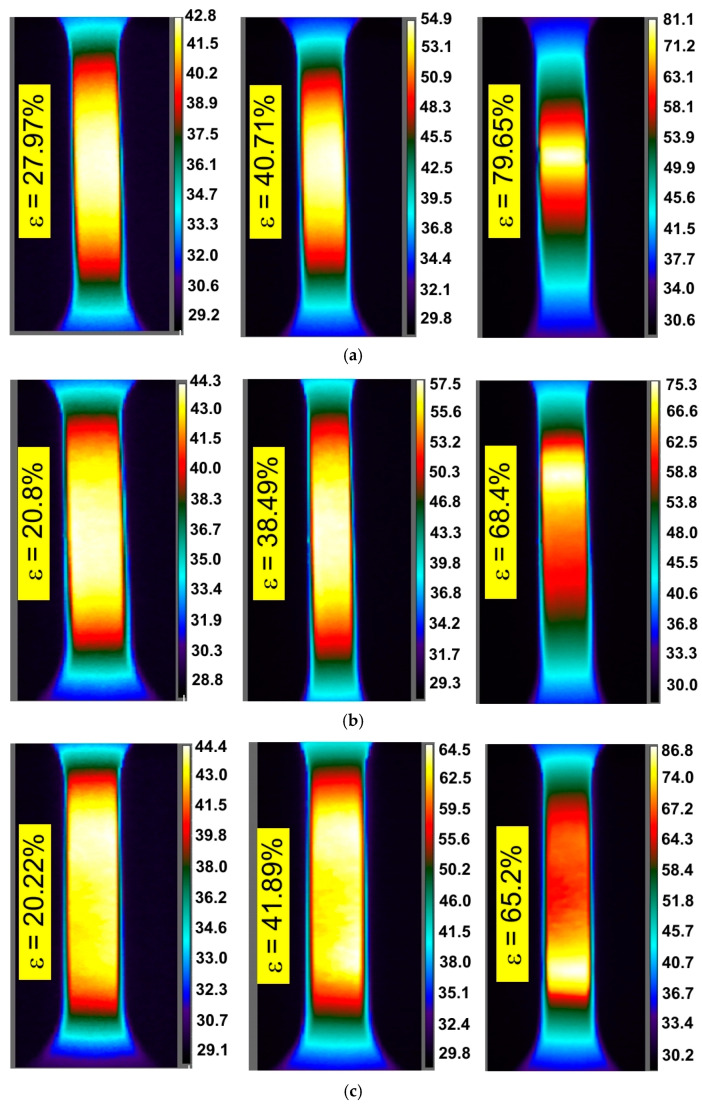
Infrared thermography images for HX specimens cut at an angle of 45° with respect to the rolling direction of the sheet metal tested at strain rates: (**a**) 0.004 s^−1^, (**b**) 0.008 s^−1^ and (**c**) 0.016 s^−1^.

**Figure 11 materials-14-02163-f011:**
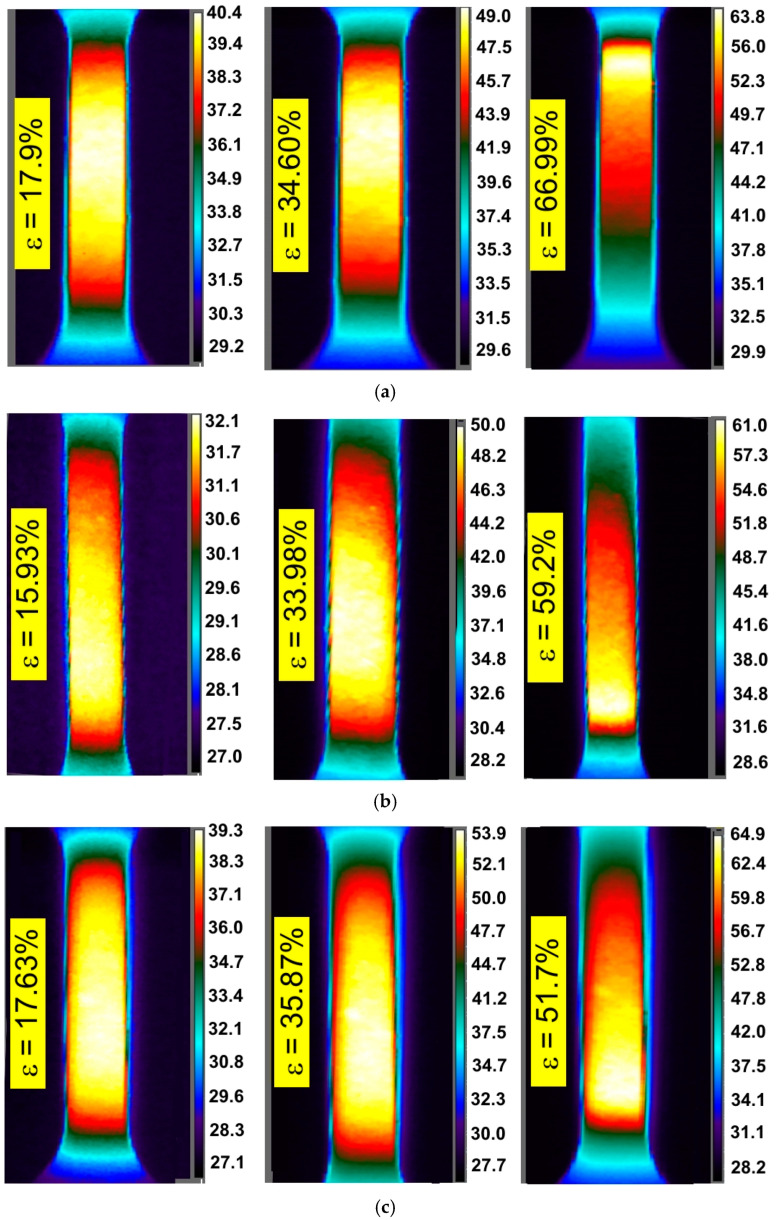
Infrared thermography images for HX specimens cut at an angle of 90° with respect to the rolling direction of the sheet metal tested at strain rates: (**a**) 0.004 s^−1^, (**b**) 0.008 s^−1^ and (**c**) 0.016 s^−1^.

**Figure 12 materials-14-02163-f012:**
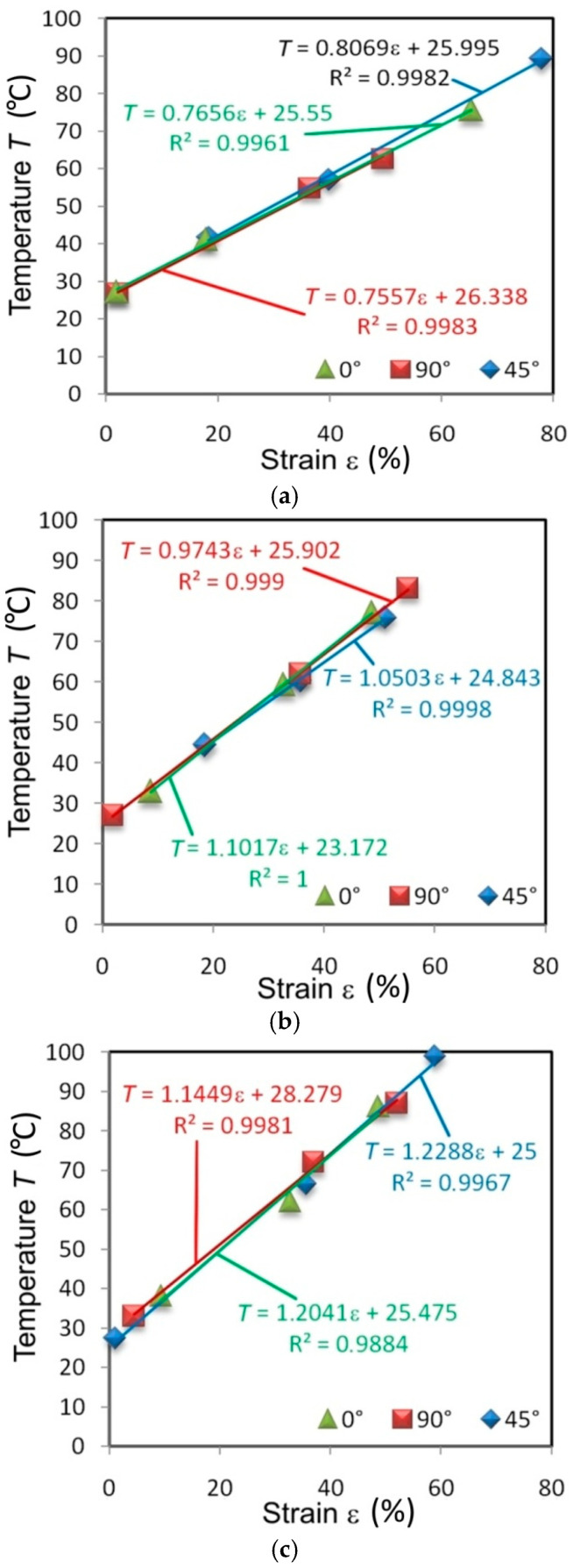
Effect of strain on the maximum temperature of I625 specimens tested at strain rates: (**a**) 0.004 s^−1^, (**b**) 0.008 s^−1^ and (**c**) 0.016 s^−1^.

**Figure 13 materials-14-02163-f013:**
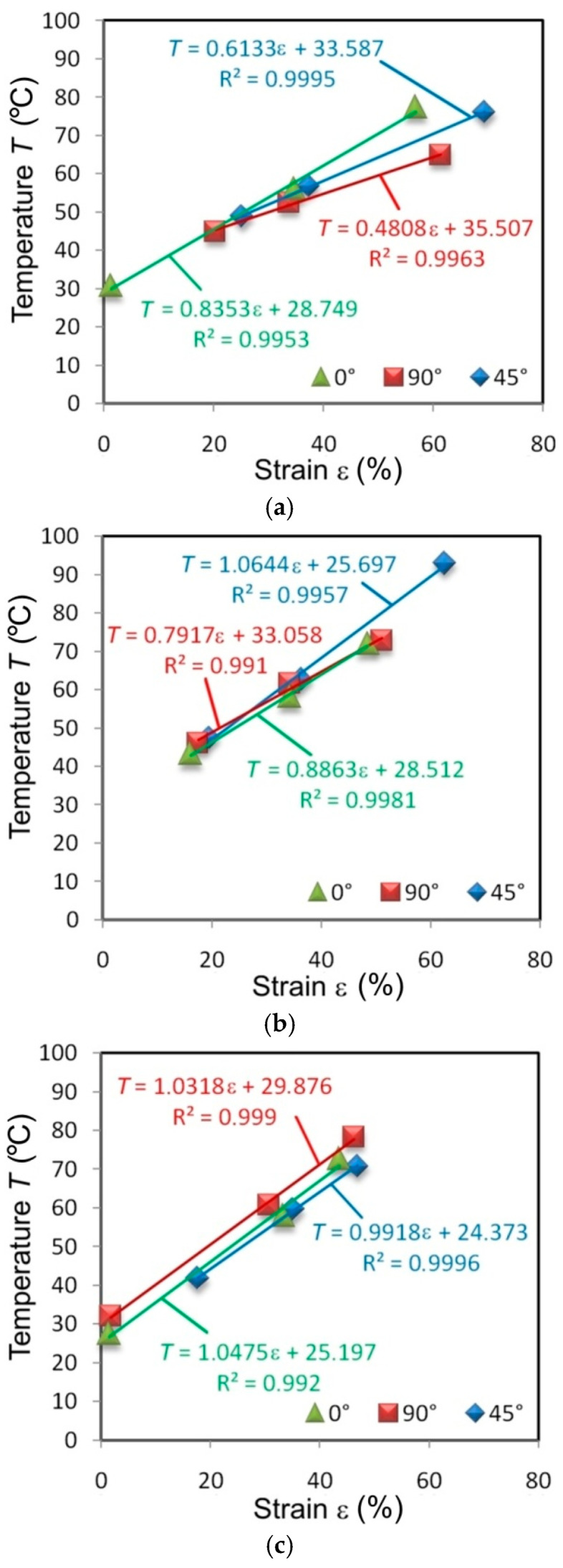
Effect of strain on the maximum temperature of I718 specimens tested at strain rates: (**a**) 0.004 s^−1^, (**b**) 0.008 s^−1^ and (**c**) 0.016 s^−1^.

**Figure 14 materials-14-02163-f014:**
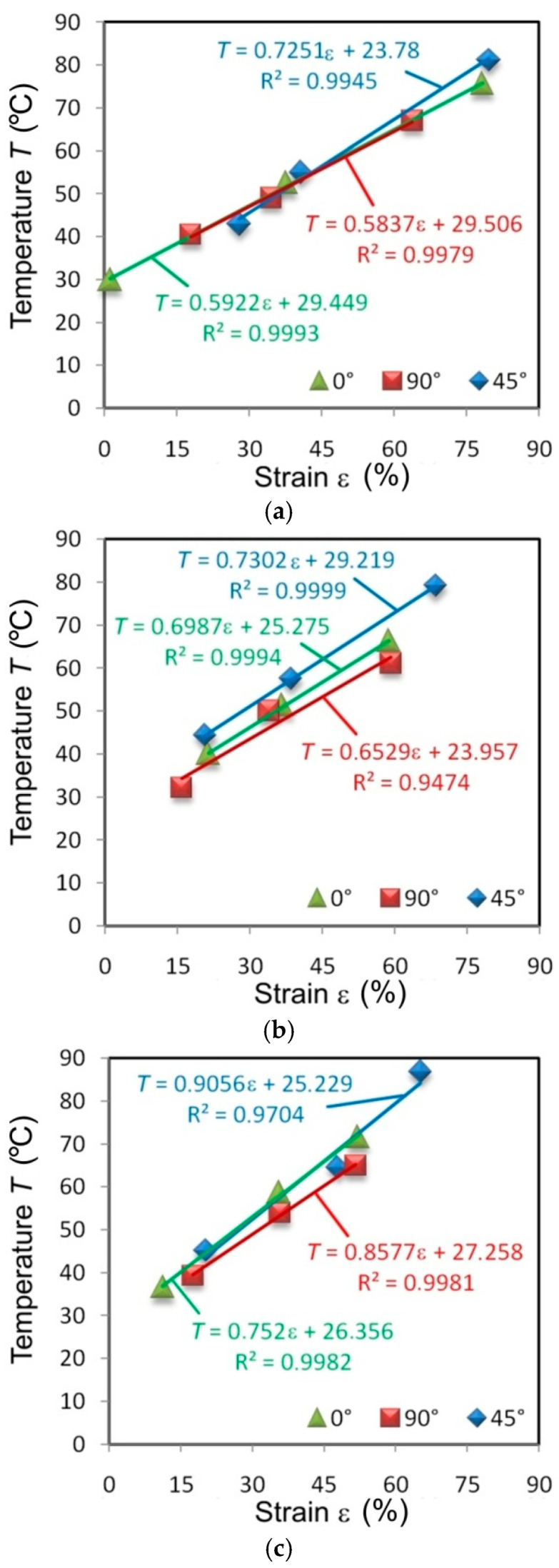
Effect of strain on the maximum temperature of HX specimens tested at strain rates: (**a**) 0.004 s^−1^, (**b**) 0.008 s^−1^ and (**c**) 0.016 s^−1^.

**Figure 15 materials-14-02163-f015:**
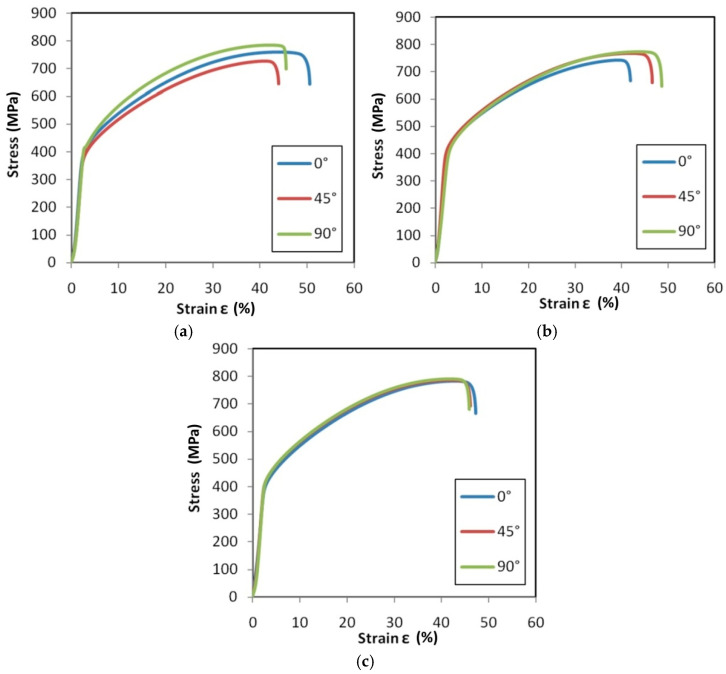
Tensile stress–strain curves for HX specimens tested at strain rates: (**a**) 0.004 s^−1^, (**b**) 0.008 s^−1^ and (**c**) 0.016 s^−1^.

**Figure 16 materials-14-02163-f016:**
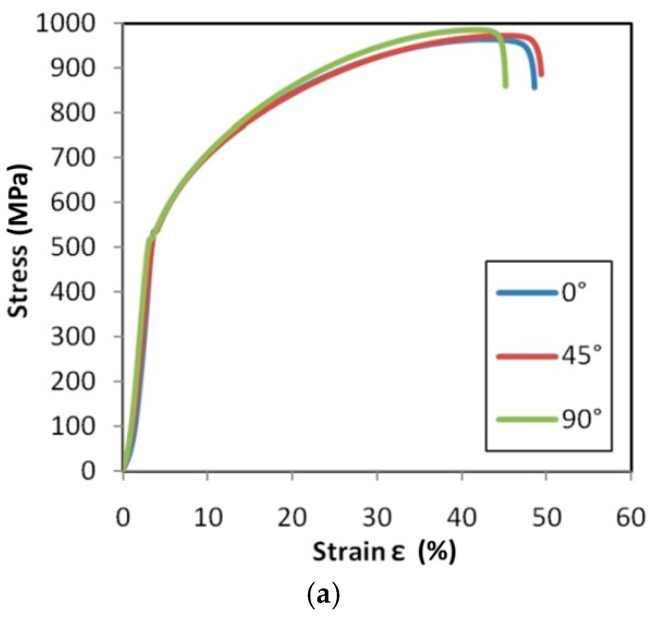
Tensile stress–strain curves for I625 specimens tested at strain rates: (**a**) 0.004 s^−1^, (**b**) 0.008 s^−1^ and (**c**) 0.016 s^−1^.

**Figure 17 materials-14-02163-f017:**
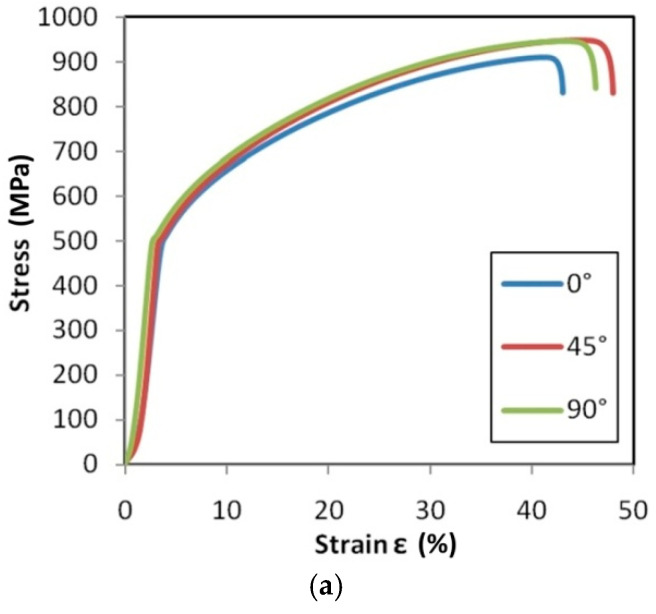
Tensile stress–strain curves for I718 specimens tested at strain rates: (**a**) 0.004 s^−1^, (**b**) 0.008 s^−1^ and (**c**) 0.016 s^−1^.

**Figure 18 materials-14-02163-f018:**
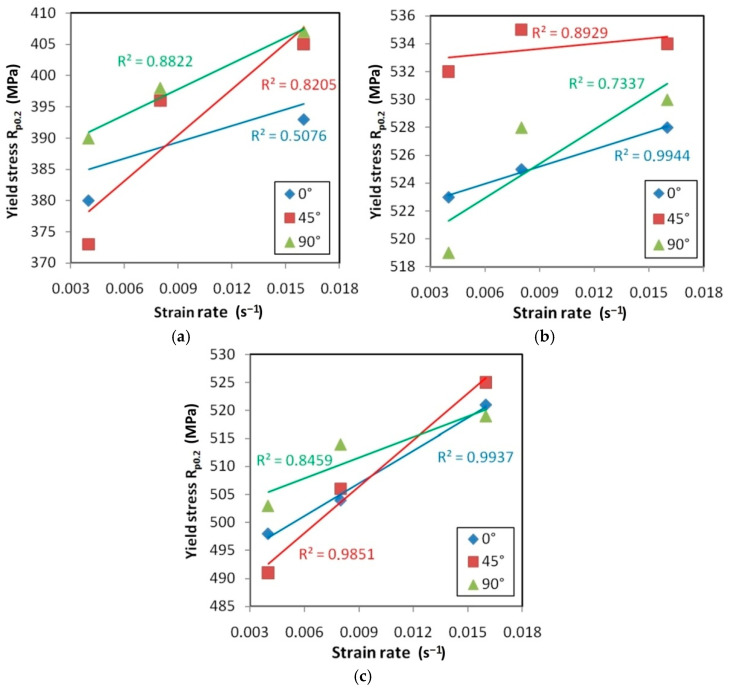
Yield stress vs. strain rate for (**a**) HX, (**b**) I625 and (**c**) I718.

**Figure 19 materials-14-02163-f019:**
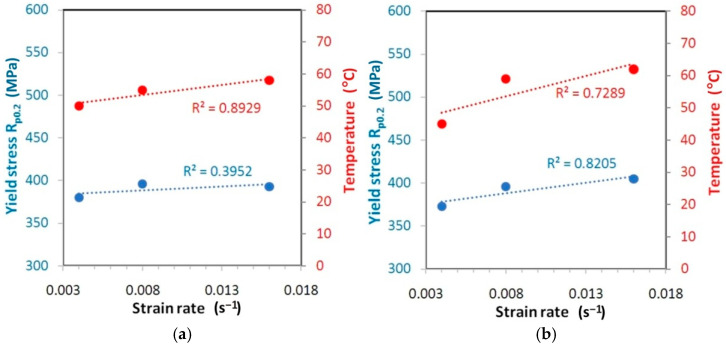
Correlation between the variation of the yield stress and measured temperature for the HX specimens cut at (**a**) 0° (ε = 35.5–37.6), (**b**) 45° (ε = 38.5–42) and (**c**) 90° (ε = 34–35.8).

**Figure 20 materials-14-02163-f020:**
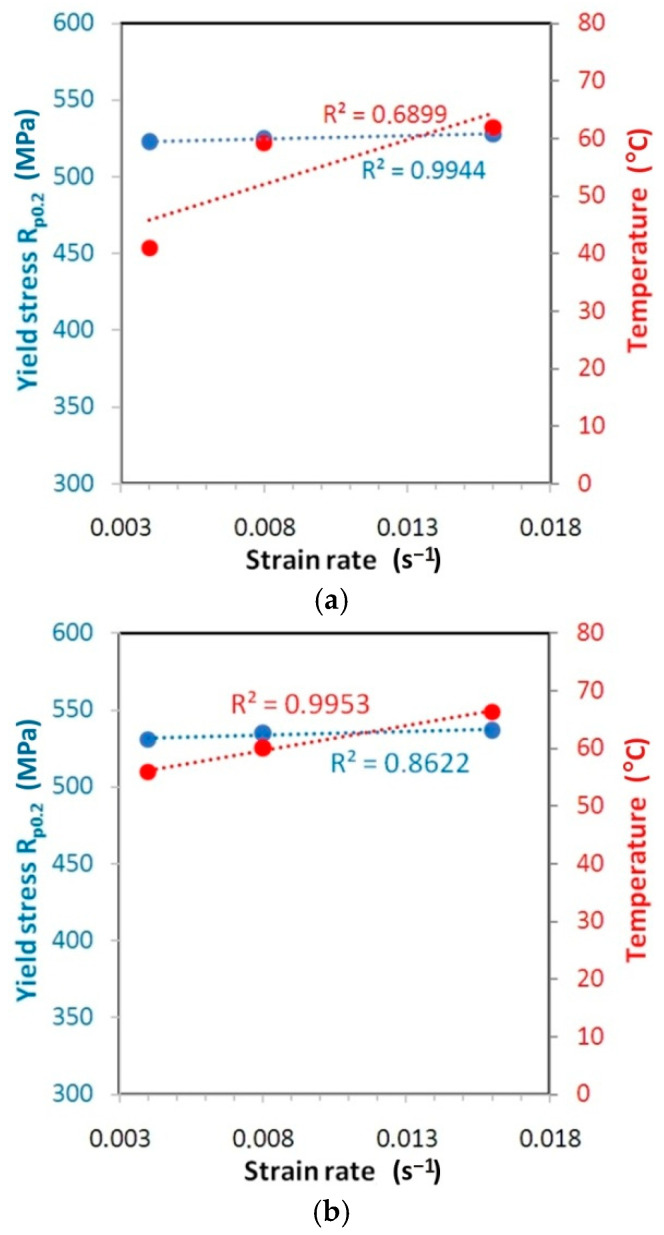
Correlation between the variation of the yield stress and measured temperature for the I625 specimens cut at (**a**) 0° (ε = 37.2–32.8), (**b**) 45° (ε = 35.6–39.9) and (**c**) 90° (ε = 35.8–37).

**Figure 21 materials-14-02163-f021:**
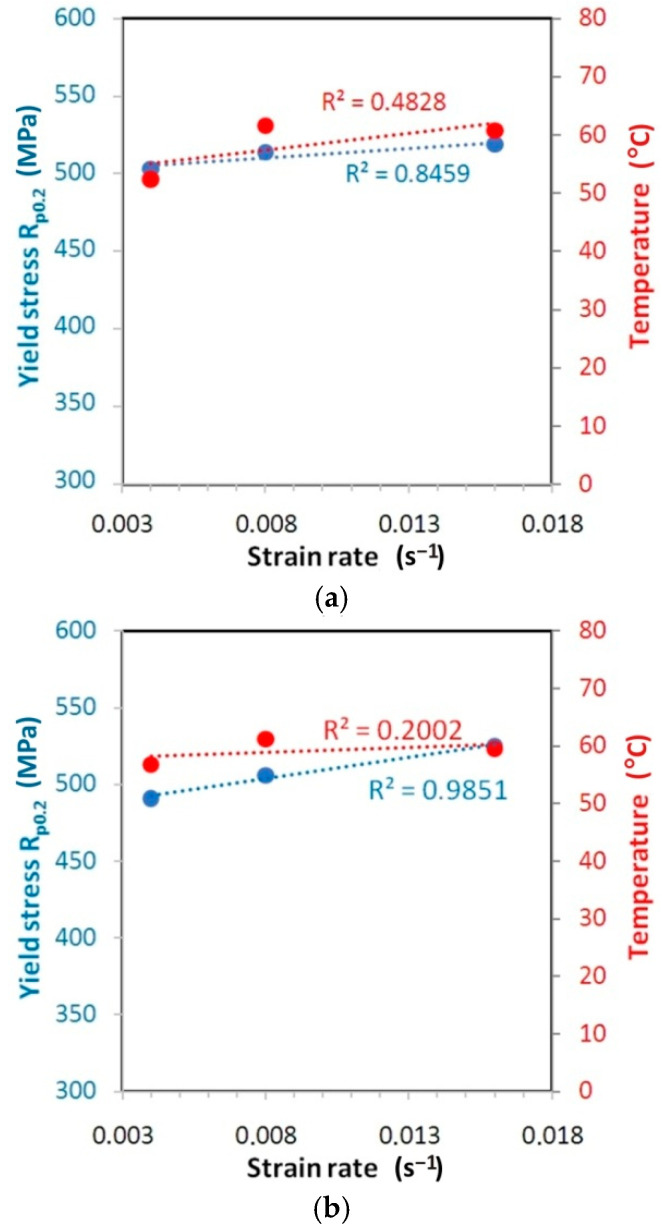
Correlation between the variation of the yield stress and measured temperature for the I718 specimens cut at (**a**) 0° (ε = 33.4–34.7), (**b**) 45° (ε = 35.1–37.3) and (**c**) 90° (ε = 30.7–34.2).

**Figure 22 materials-14-02163-f022:**
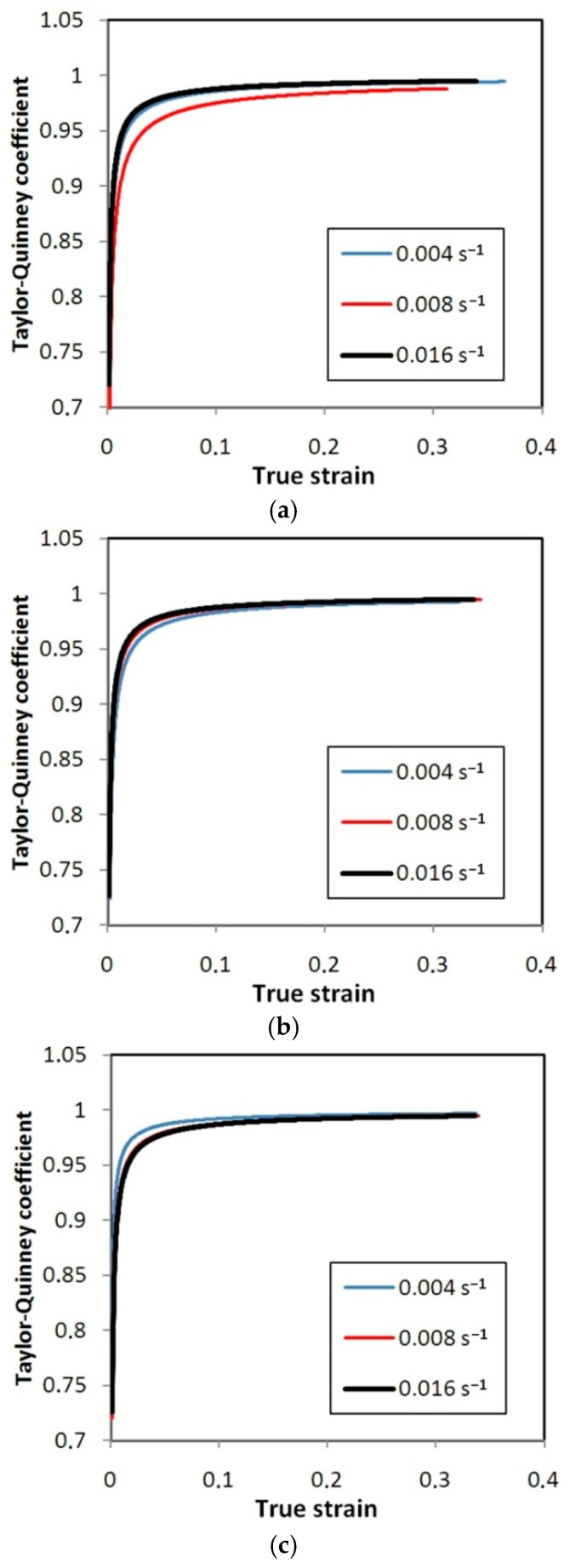
Variation of the Taylor–Quinney coefficient vs. true strain for HX specimens oriented at: (**a**) 0°, (**b**) 45° and (**c**) 90°.

**Figure 23 materials-14-02163-f023:**
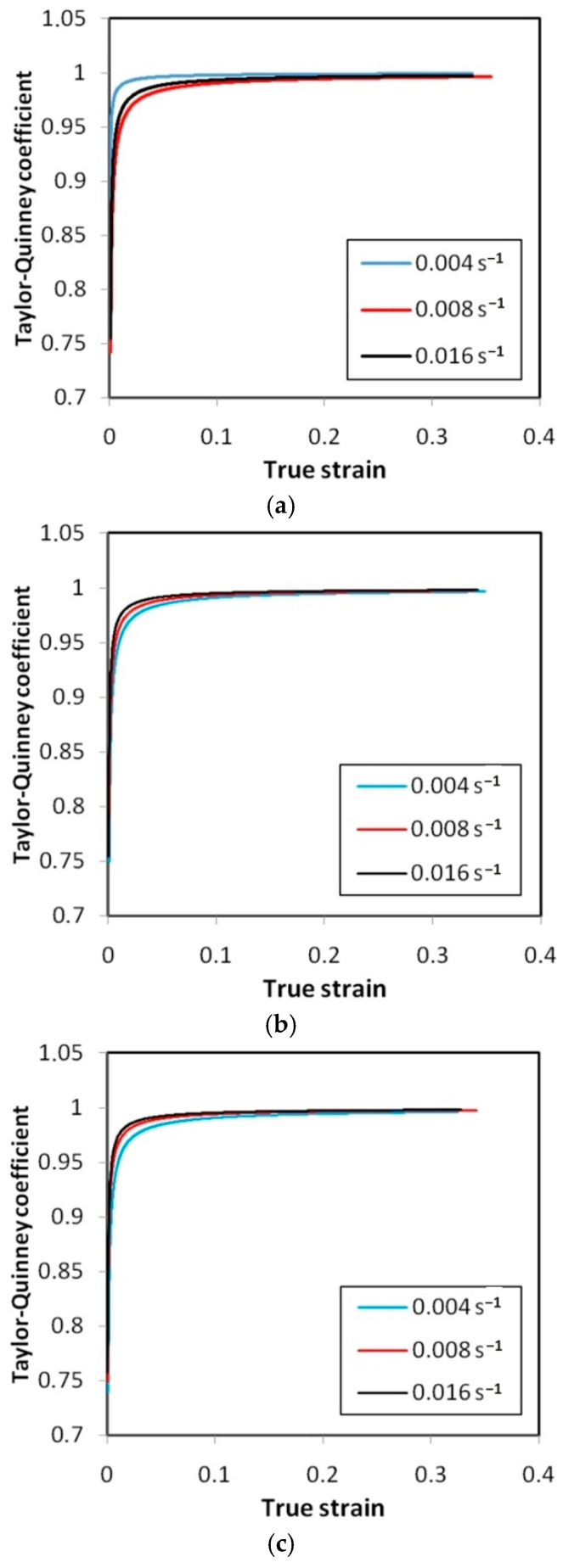
Variation of the Taylor–Quinney coefficient vs. true strain for I625 specimens oriented at: (**a**) 0°, (**b**) 45° and (**c**) 90°.

**Figure 24 materials-14-02163-f024:**
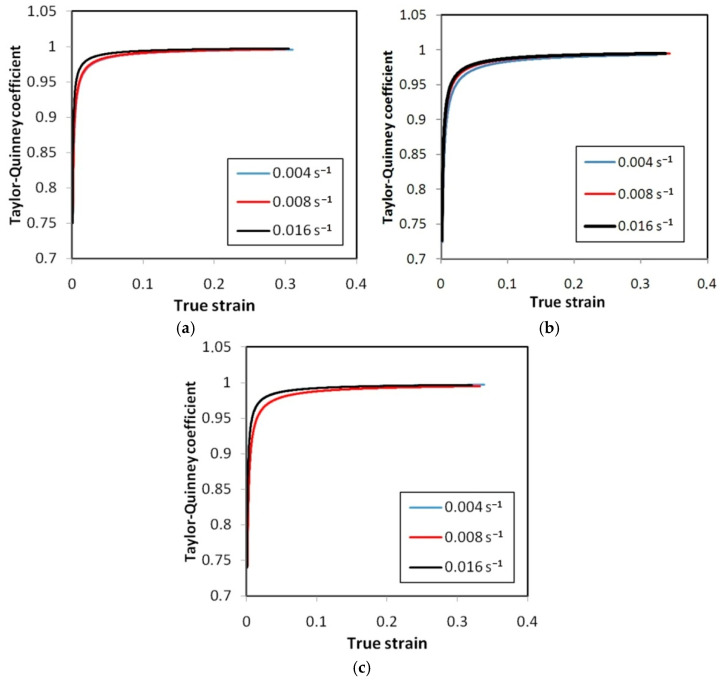
Variation of the Taylor–Quinney coefficient vs. true strain for I718 specimens oriented at: (**a**) 0°, (**b**) 45° and (**c**) 90°.

**Table 1 materials-14-02163-t001:** Chemical composition of Inconel alloys (wt.%) [32,33,34].

Material	Ni	Cr	Fe	Mo	Nb	C	Mn	Si	P	S	Al	Ti	Co	B	Cu	Ta
I625	58 (min)	20–23	5 (max)	8–10	3.15–4.15 *	0.1 (max)	0.5 (max)	0.5 (max)	0.015 (max)	0.015 (max)	0.4 (max)	0.4 (max)	1.0 (max)	–	–	–
HX	balance	20.5–23.0	17.0–20.0	8.0–10.0	–	0.05–0.15	1.0 (max)	1.0 (max)	0.04 (max)	0.03 (max)	–	0.2–1.0	0.5–2.5	–	–	–
I718	50–55	17–21	balance	2.80–3.30	4.75–5.50	0.08 (max)	0.35 (max)	0.35 (max)	0.015 (max)	0.015 (max)	0.20–0.80	0.65–1.15	1.00 (max)	0.006 (max)	0.30 (max)	0.05 (max)

*—plus Ta.

## Data Availability

The data presented in this study are available on request from the corresponding author.

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
