# Peer review of "Coupled Thermomechanical Response Measurement of Deformation of Nickel-Based Superalloys Using Full-Field Digital Image Correlation and Infrared Thermography"

_materials, 2021, doi:10.3390/ma14092163_

Round 1

Reviewer 1 Report

This paper describes measurement of thermomechanical response during tensile deformation of nickel-based superalloys using full-field digital image correlation and infrared thermography. The experimental results are interesting. My comments are as follows.

1.Stress-strain curves of the I625, HX, and I718 at strain-rates of 0.004, 0.008, and 0.016 s-1 are required. Was the linear strain-rate dependence of the yield stress observed in this strain-rate range? Can the authors find correlation between the variation of the yield stress and measured temperature increase of the alloys?

2. Line 192: at 30 C => at 30 oC

3. What would the authors like to discuss by using Eq. (5) and (6) on their results? Please describe clearly.

4. It is difficult to distinguish differences in the values of slope of the straight lines in Fig. 8, 9, and 10 at a glance. Showing the relationships between the values of slope of the straight lines and strain rates is helpful for readers. 

Reviewer 2 Report

The paper presents a coupled thermomechanical response measurement of deformation of Nickel-based superalloys using full-field digital image correlation and infrared thermography. The authors provide an adequate introduction to the study. The abstract does not adequately highlight the centered objectives, goals and improvements that the study brings. The experimental methods are adequately described but the results lack an effective contribution to the part of the strain measurements carried out. They are commented on the basis of the data acquired during the tests but are not explicitly reported. In addition, highlighting the novelty that the  introduced methodology of study and the results obtained is necessary. A review of formatting and English is also necessary.

Reviewer 3 Report

1. lines 38 instead of “...in the metallic material as phase changes, permanent microstructural changes and internal defects.”it seems to be better to use “...in the metallic material as phase changes, permanent microstructural changes including internal defects formation .” It is desirable to use specific references for phase transformations and structure evolution, which the authors mentioned above.

2. line 56 “electromagnetic radiation” instead of “magnetic radiation”.

3. line 83. “thermography” instead of “thermo-graphy».

4. Fig3,7. Some images posses a slight tilt. Why?

5. Many numbers in the figures are too small to read.

6. Cloud charts should be explained more thoroughly so that the reader can clearly understand what are the sense an the real reason to use authors method.

Reviewer 4 Report

Dear Authors,

I have read your paper "Coupled Thermomechanical Response Measurement of Deformation of Nickel-based Superalloys Using Full-Field Digital Image Correlation and Infrared Thermography" carefully. 

This paper describes the thermo-mechanical response measurements of the Inconel alloys during uniaxial tensile tests. 

The paper is easy to read.

Methods are properly described, so that other research groups may reproduce them.

The paper is interesting. However, it requires few corrections.

  1. Figure 10,c. Is there mistake? 
  2. Please specifically discuss the advantages of your work. Some parts of your conclusion can be writing in the discussion section. Is there the effect of the diameter of the wire on the process? Can you add standart tensile strain-stress curves?

 The paper can be accepted for publication after minor improvements.

Round 2

Reviewer 2 Report

After the revisions made by the authors, the paper's objectives are clearer as well as the results obtained. A major improvement has been made in the discussion part of the results. The paaper is adequate for publication.